# State of the Art: ctDNA in Upper Gastrointestinal Malignancies

**DOI:** 10.3390/cancers15051379

**Published:** 2023-02-21

**Authors:** Ibone Labiano, Ana Elsa Huerta, Virginia Arrazubi, Irene Hernandez-Garcia, Elena Mata, David Gomez, Hugo Arasanz, Ruth Vera, Maria Alsina

**Affiliations:** 1Oncobiona Group, Navarrabiomed-Instituto de Investigación Sanitaria de Navarra (IdiSNA), Irunlarrea 3, 31008 Pamplona, Spain; 2Medical Oncology Department, Hospital Universitario de Navarra (HUN), Irunlarrea 3, 31008 Pamplona, Spain

**Keywords:** ctDNA, gastrointestinal tumors, liquid biopsy, cancer, precision medicine

## Abstract

**Simple Summary:**

We present an exhaustive review of the literature that has evaluated the role of ctDNA analysis in upper gastrointestinal tumors, including gastroesophageal adenocarcinoma (GEC), biliary tract cancer (BTC) and pancreatic ductal adenocarcinoma (PADC). We describe the implications of ctDNA from early diagnosis to molecular characterization and follow-up of tumor genomic evolution, from a current point of view and debating strengths and weaknesses.

**Abstract:**

Circulating tumor DNA (ctDNA) has emerged as a promising non-invasive source to characterize genetic alterations related to the tumor. Upper gastrointestinal cancers, including gastroesophageal adenocarcinoma (GEC), biliary tract cancer (BTC) and pancreatic ductal adenocarcinoma (PADC) are poor prognostic malignancies, usually diagnosed at advanced stages when no longer amenable to surgical resection and show a poor prognosis even for resected patients. In this sense, ctDNA has emerged as a promising non-invasive tool with different applications, from early diagnosis to molecular characterization and follow-up of tumor genomic evolution. In this manuscript, novel advances in the field of ctDNA analysis in upper gastrointestinal tumors are presented and discussed. Overall, ctDNA analyses can help in early diagnosis, outperforming current diagnostic approaches. Detection of ctDNA prior to surgery or active treatment is also a prognostic marker that associates with worse survival, while ctDNA detection after surgery is indicative of minimal residual disease, anticipating in some cases the imaging-based detection of progression. In the advanced setting, ctDNA analyses characterize the genetic landscape of the tumor and identify patients for targeted-therapy approaches, and studies show variable concordance levels with tissue-based genetic testing. In this line, several studies also show that ctDNA serves to follow responses to active therapy, especially in targeted approaches, where it can detect multiple resistance mechanisms. Unfortunately, current studies are still limited and observational. Future prospective multi-center and interventional studies, carefully designed to assess the value of ctDNA to help clinical decision-making, will shed light on the real applicability of ctDNA in upper gastrointestinal tumor management. This manuscript presents a review of the evidence available in this field up to date.

## 1. Introduction

Currently, histotype diagnosis on tumor tissue is the gold standard for tumor characterization and first therapeutic approach guidance. Moreover, tissue analyses are needed to assess the local tumor microenvironment, for example to know the programmed death ligand-1 (PD-L1) status or the tumor-infiltrating lymphocyte (TIL) characterization. For a further tumor genomic characterization, tissue genomic testing in tissue is the gold standard to identify genomic alterations, assess their prognostic value, select patients for matched targeted therapy and monitor their responses. However, tissue-based genomic testing presents important limitations; mainly, it requires an invasive procedure, tissue may not be available depending on tumor location, and it may not recover full tumor heterogeneity [1,2]. In this scenario, genomic testing in ctDNA arises as a promising strategy to overcome these pitfalls. Importantly, as aforementioned, ctDNA analysis overcomes the limitations of tissue testing in terms of polyclonal heterogeneity and tissue availability; moreover, due to its minimally invasive nature, it allows for repetitive testing, providing real-time information on the tumor biology [3].

Liquid biopsy is the concept of analyzing biologic material derived from tumor cells into bodily fluids. Among different biologic materials found in liquid biopsies, cell free DNA (cfDNA) is fragmented DNA that can originate from, potentially, any cell of the organism. Circulating tumor DNA (ctDNA), is the fraction of cfDNA released from tumor cells and may reflect the genomic as well as epigenomic landscape of the tumor [4]. 

## 2. Technical Aspects of ctDNA Analysis

Detailed technical aspects of ctDNA analysis are beyond the scope of this manuscript, and only some important considerations will be highlighted; for further information, readers are referred to recent publications in the field [5,6]. Two main approaches are used for the detection of genetic alterations in ctDNA. Advanced polymerase chain reaction (PCR)-based techniques are allele-specific approaches that interrogate previously known mutations with high sensitivity [5]. Alternatively, next-generation sequencing (NGS)-based techniques analyze several alterations in a unique experiment, although with lower sensitivity, and they can detect de novo alterations including copy number variations and rearrangements. NGS analyses can focus on a panel of selected genes or perform whole-exome or genome sequencing [7,8]. Each approach can be complementary and useful depending on the context and purpose of the assay. PCR-based techniques may be more suitable for the detection of a common, previously identified mutation, while NGS techniques cover the entire molecular characterization and over-time tumor clonality of patients receiving targeted therapy [5,6]. Finally, low-coverage whole-exome or genome sequencing have been employed for the detection of copy number variations (CNVs) [9]. 

An important technical limitation of ctDNA analysis relates to the low amounts of DNA present in the sample in some settings, preventing detection of present alterations with current methods. In line with this, detection of CNVs and genomic rearrangements in ctDNA with current technologies is also considered suboptimal [5,10]. Moreover, some tumors, such as primary brain tumors, renal, prostate and thyroid cancers, are regarded as non-shedders and may not be amenable to ctDNA analysis [11]. Finally, blood cells accumulate somatic mutations with age in a process named clonal hematopoiesis of indeterminate potential, which represents a source of false positives in ctDNA analysis [12]. 

Figure 1 presents the characteristics of genomic testing in tissue and ctDNA of liquid biopsy. In line with this, a recent study that included 1021 patients with different solid tumors revealed that ctDNA sequencing detected a considerable amount of targetable alterations not present in tissue (9% ESCAT-tier I/II, 14% ESCAT-tier III, and 6% ESCAT-tier IV), while in 7% of the patients the alterations were only found in tissue. This study suggests that ctDNA genetic testing can complement and even replace tissue testing [2]. Indeed, the ESMO has recently published their recommendations for genetic testing in ctDNA in several solid tumors [10], essentially when rapid results are needed and tissue is unavailable. 

## 3. Clinical Applications of ctDNA Analysis in GEC, BTC and PDAC

The analysis of ctDNA and its clinical applicability is most advanced in lung, breast and colon cancer. Indeed, accumulating evidence supports the clinical usefulness of this approach at different time points along disease progression, including early diagnosis, detection of minimal residual disease (MRD) after surgery, molecular characterization for selection of targeted therapy, patient prognosis, and monitoring of treatment response and emergence of resistance to targeted therapy in the advanced setting [2,3,6,13,14]. In the present review, recent advances in the applications and usefulness of ctDNA in upper gastrointestinal cancers, namely, gastroesophageal adenocarcinoma (GEC), biliary tract cancer (BTC) and pancreatic ductal adenocarcinoma (PDAC) will be presented and discussed. Of note, these malignancies represent hard-to-treat cancers with poor prognosis, which would potentially benefit from the implementation of ctDNA analysis in different clinical contexts [6,15,16].

Importantly, GEC, BTC and PDAC are usually diagnosed at advanced stages, when no longer amenable to surgical resection. In this context, it is critical to characterize the genomic landscape of the tumor, with the aim of identifying targetable alterations and selecting patients who would potentially benefit from matched targeted therapies [16]. Overall, targeted therapy approaches improve patient survival compared to conventional chemotherapy [17,18,19,20,21]. Nevertheless, distinguishing the real clinical value of a given genomic alteration remains challenging [22,23]. Along this line, important scientific societies have proposed guidelines in order to help in the interpretation of the potential clinical utility of genomic alterations in solid tumors [24,25,26]. The European Society of Medical Oncology (ESMO) proposed a classification system for genomic alterations based on their clinical evidence of actionability [27]. The ESMO Scale of Actionability of molecular Targets (ESCAT) establishes six progressive levels according to the existence of matched drugs for a given genomic alteration and their reported clinical benefit: from tier I, in which alteration-drug match is associated with improved outcome in clinical trials, to tier IV, where only preclinical evidence of actionability is reported. Tier IV and X mean a lack of clinical benefit or evidence of actionability, respectively [27]. Interestingly, in a study assessing the clinical applicability of this scale that included 552 patients with different solid tumors in the advanced setting, 67% of the patients showed at least one actionable genetic alteration, and 27% of those were treated with a matched therapy. Importantly, patients harboring alterations of a stronger clinical evidence (tiers I and II) and treated with a matched drug showed longer progression-free survival (PFS) compared to those harboring alterations with a weaker clinical meaning (tiers III and IV) [22]. In this scenario, the ESMO has recently reviewed the genetic landscape and recommendations for genetic testing in several tumors in the advanced setting [28]. Focusing on gastrointestinal malignancies, the ESMO recommends genetic testing of specific alterations in GEC (*ERBB2* amplifications, MSI-H, and *NTRK* 1,2,3 fusions), BTC (*IDH1* mutations, *FGFR2* fusions, MSI-H, and *NTRK* 1,2,3 fusions) and PDAC (MSI-H and *NTRK* 1,2,3 fusions) [28]. Similarly, the American Society of Clinical Oncology (ASCO) has recently proposed their Provisional Clinical Opinion (PCO), a panel of recommendations addressing specific clinical questions related to genomic testing in solid tumors. Overall, genomic testing is recommended for tumors with biomarker-based therapeutic indications and multigene testing when more than one targeted therapeutic indication is available [29].

The evidence of the clinical usefulness of ctDNA in GEC, BTC and PDAC is still limited, and prospective multi-center studies are needed in order to shed light on its real clinical applicability in the next future. In the following lines, pan-cancer studies including GEC, BTC and PDAC patients as well as small studies focused on each of the malignancies will be discussed.

Analyzing methylation profiles in ctDNA has been proposed as a valuable tool for early cancer diagnosis, as epigenetic changes, mainly methylation, could precede genetic changes. Pan-cancer studies including thousands of patients with different malignancies have proposed cfDNA concentrations or methylation panels with high diagnostic capacities and the ability to distinguish primary tumor sites [11,30,31,32,33]. 

In the advanced setting, the SCRUM-Japan GOZILA study assessed ctDNA-based genetic testing utility for targeted-therapy trial enrolment in patients with advanced gastrointestinal cancers (including colorectal carcinoma, GEC, squamous cell carcinoma, BTC and PDAC) and compared its performance to tissue genetic testing. Importantly, ctDNA-based genetic testing shortened enrolment time and improved enrolment rate without influencing response rates [34]. ctDNA could also be used to predict treatment response and monitor relapses. In two proof-of-concept studies assessing ctDNA levels in pan-cancer patients treated with immunotherapy, ctDNA analysis had a prognostic value and could predict responses [35,36]. Finally, ctDNA analysis has the capability to monitor polyclonal resistance mechanisms arising in targeted therapy-treated patients. A study including 42 patients with advanced gastrointestinal tumors that developed resistance upon targeted therapy identified novel resistance alterations not found in tissue biopsies for 78% of the cases. Importantly, among the 23 patients analyzed at tissue and ctDNA level, tissue biopsy identified multiple resistance mechanisms in 9% of patients, while by ctDNA this percentage rose to 40% [37]. Table 1 summarizes key findings and study designs of selected studies assessing the role of ctDNA in different types of cancer.

## 4. Gastroesophageal Cancer (GEC)

Gastric cancer represents a global healthcare challenge. With an estimated 1,089,103 new cases and 768,793 new deaths in 2020, this tumor type ranks fifth in incidence and fourth in mortality. Further, the gastroesophagogastric junction adenocarcinoma incidence is increasing [38]. GEC, referring to both tumors together, has been historically referred as one unique entity, although it comprises four different molecular subtypes [39,40,41] that recognize the well-described interpatient heterogeneity, a major cause of failure of phase II-III clinical trials with targeted therapies [20,42]. Additionally, variations within the same tumor (intrapatient-intratumoral-heterogeneity) have also been described, from a spatial and temporal perception [43,44,45]. In this regard, especially in GEC, ctDNA appears as a novel approach to overcome this obstacle.

### 4.1. Screening and Diagnosis

Except in high-risk East Asian countries (China, Japan and South Korea), there are no population GEC-screening strategies, and most GEC patients are diagnosed with an advanced disease [46]. The first approach to tumor diagnosis and characterization is based on the tissue histotype, and the performance of genetic testing adds value for prognostic and treatment purposes. Multiple (5–8) biopsies should be carried out to ensure enough material for a first and necessary histological and molecular interpretation [47,48]. Sequencing the tumor tissue also requires an invasive procedure in order to obtain enough tumor cells. In this regard, ctDNA analyses could cover both limitations, i.e., facilitate the invasive procedures and overcome the spatial heterogeneity. Some studies have shown that ctDNA can be detected in patients with GEC more often than in patients with benign pathologies of the stomach or in healthy controls [33,49], thus proposing it as a potential non-invasive tool for massive screening. Furthermore, and as mentioned before, the use of DNA methylation patterns for screening purposes has also been also investigated in GEC as in other tumors, although the sensitivity reported is low for early tumor stages [32]. Prospective studies evaluating the potential use of ctDNA for early diagnosis of GEC are currently ongoing in South Korea (NCT04665687), US (NCT04241796) and UK [50].

### 4.2. MRD and Recurrence Monitoring

For patients with potentially resectable GEC, standard treatment includes peri-operative combination chemotherapy and surgery in Western countries [51]. Adjuvant chemotherapy is indicated regardless of the tumor response to the neoadjuvant treatment, and there is a lack of reliable programs to monitor recurrence [46]. The use of ctDNA to identify GEC patients at risk of recurrence has been limited to small cohorts, varying assays and time points. Different studies have confirmed the association of pre-operative ctDNA levels with different tumor stages [11,32], and how the detection of ctDNA in the immediate post-operative period correlates with eventual recurrences [33,52,53,54,55,56,57,58]. Notably, a subanalysis of 50 patients included in the phase III CRITICS study, which randomized patients to receive pre-operative chemotherapy and surgery plus post-operative chemotherapy vs. post-operative chemoradiotherapy, demonstrated that the presence of ctDNA within nine weeks of surgery predicted recurrence [59]. Additionally, a retrospective analysis of real-world data including 295 patients showed that ctDNA detection at any time point after surgery or during the surveillance period was associated with shorter recurrence-free survival [60]. Currently, there are different ongoing studies prospectively assessing the value of the MRD-ctDNA detection as a key tool for deciding on the adjuvant treatment indication, in patients with GEC (NTC04510285, NCT02674373).

### 4.3. Metastatic Disease Monitoring

Advanced GEC patients show a very poor prognosis with a 5-year relative survival rate of 6%. Treatment with chemotherapy has been shown to improve overall survival (OS) and quality of life, compared to best supportive care alone [61]. For the time being, only two tissue biomarker subpopulations have been identified for targeted treatment: the human epidermal growth factor (HER2)-positive and PD-L1-positive subpopulations, with potential benefit from the addition of trastuzumab and anti-PD1 agents to the first-chemotherapy line of treatment, respectively [62,63,64]. As in other tumors, ctDNA detection and sequencing in GEC patients could provide valuable genetic information and allow to follow up on the tumor genomic evolution without the need for serial tissue biopsies. Again, one should recognize the intrinsic limitations related to the analyses of circulating genomics but not local peri-tumor protein expression. One of the largest retrospective studies of GEC patients demonstrated how ctDNA can potentially recover the temporal and spatial molecular heterogeneity, and how temporal changes in the tumor somatic variant allelic frequency (VAF) correlate with prognosis in patients receiving chemotherapy and immunotherapy [52]. These findings have been corroborated by other smaller studies [65], including the association of basal ctDNA levels with prognosis [66], and how ctDNA changes over time correlate with response to different treatments [66,67,68]. 

When considering the HER2-positive subpopulation of GEC, the detection of *HER2* amplification in ctDNA has been proposed as an optimal tool to overcome the challenge of tumor temporal and spatial heterogeneity, thus being able to predict and monitor responses to anti-HER2 therapies, and to inform about possible mechanisms of resistance [45,52,69,70,71,72]. Ongoing observational and interventional studies would probably validate these findings (NTC04520295, NTC03409848).

Blood samples from Epstein Barr Virus (EBV)-positive GEC tumors have been analyzed. Although the plasma EBV-DNA load has been identified only in half of EBV-(tissue)-positive cases, it could correlate with response to treatment [73]. Furthermore, detection of *FGFR2b* amplification in ctDNA has been partially associated with responses to anti-FGFR2b targeted therapies [74,75], but not *EGFR* amplifications [76].

Concerning historical approaches to individual molecular alterations focusing on single targeted therapies, sequencing techniques could lead to a multiplex approach to define the best personalized treatment algorithm, and ctDNA analysis arises as the best approach to follow up on clonal tumor evolution. The VIKTORY trial [77] was an umbrella trial conducted in South Korea, which assigned patients with metastatic GEC to one of 10 phase II molecularly-driven clinical trials for a second-line treatment, depending on eight different tumor biomarkers on tissue-based sequencing, although ctDNA was analyzed at baseline and longitudinally. Patients receiving the biomarker-selected therapy presented prolonged PFS and OS compared with patients receiving conventional treatment, and reduction of ctDNA levels correlated with response to treatment. The PANGEA trial [45] considered the first three lines of treatment, with optimally sequenced chemotherapy plus different monoclonal antibodies, depending on the molecular findings in tissue and ctDNA. Genomic discordance was observed in tissue between primary and metastatic tumors in 35% of patients, with better concordance when comparing results of metastatic tissue and ctDNA. Additionally, tumor changes after a first- and second-targeted line of treatment was identified in 50% of the treated patients. Meeting its primary endpoint of OS, this study again confirms the spatial and temporal heterogeneity of GEC and demonstrates the feasibility and efficacy of molecular approaches. Finally, and from a country approximation, the already mentioned GOZILA initiative [78] performed comprehensive ctDNA sequencing to rapidly screen cancer patients for trial eligibility, including GEC patients; the authors demonstrated that massive ctDNA genotyping unveils the presence of rare molecular targetable alterations in these patients, including tumors harboring neurotropic receptor tyrosine kinase 1 (*NTRK1*) fusions. The results of these trials highlight the potential clinical utility of ctDNA analysis in selecting patients for personalized treatment. The incorporation of ctDNA analysis as a complement in the comprehensive tumor characterization in the majority of clinical trials in patients with GEC will confirm its real value. Table 2 summarizes key findings and study designs of selected studies assessing the role of ctDNA in GEC. 

## 5. Biliary Tract Cancer (BTC)

BTC is a group of heterogeneous malignancies including mainly intra- and extrahepatic cholangiocarcinomas (CCA), as well as gallbladder cancer (GBC). The incidence of these tumors is relatively low in western countries, but significantly higher in certain geographic areas such as China and Thailand for CCA, and Chile for GBC. Moreover, the incidence of intrahepatic CCA is increasing worldwide [38,79]. When possible, surgery is the unique option of cure, although tumor recurrences are frequent. Even though the majority of patients are diagnosed in advanced stages and prognosis remains dismal, identification of distinct patient subgroups harboring unique molecular alterations with corresponding targeted therapies is improving the treatment paradigm of these patients [79,80].

### 5.1. Screening and Early Diagnosis

Early diagnosis of BTC is challenging. BTC usually present with unspecific symptoms and can be confused with benign biliary disorders that may also cause biliary stenosis {Valle, 2021, Biliary tract cancer}. In this regard, cfDNA concentration, CNV scores and/or methylation scores analyzed on ctDNA are increased in patients with CCA and/or GBC and serve to distinguish them from healthy controls and patients with benign biliary lesions [81,82,83,84]. Interestingly, in the context of BTC and PDAC, bile arises as a novel source for ctDNA analysis. Several studies have assessed ctDNA in bile collected after endoscopic biliary drainage for early diagnosis and tumor molecular characterization. Overall, ctDNA found in bile is characterized by larger fragments and shows a better correlation with tumor tissue than plasma [85,86,87,88,89]. Interestingly, bile ctDNA shows promising diagnostic capacities that outperform current diagnostic strategies [86,90]. However, the procedure to obtain bile is invasive, which prevents its use for repetitive testing. A meta-analysis comparing the diagnostic efficacy of cfDNA analysis in plasma and bile found that ctDNA detection in blood was more sensible than in bile [91]. Nevertheless, the studies included in this meta-analysis measured different cfDNA or ctDNA-related characteristics and were not homogenous in terms of the characteristics of the patients included. 

### 5.2. Metastatic Disease Monitoring

As previously mentioned, median OS of patients with advanced disease is poor, and treatment with up to two lines of chemotherapy has shown modest efficacy [79,92]. Integration of NGS techniques identifying distinct genomic alterations that underlie disease progression [93] has accelerated the treatment paradigm of BTC. Patients with intrahepatic CCA bear genetic alterations in *FGFR2* and *IDH1* mutations and benefit from matched targeted therapy [94], while in patients with extrahepatic CCA and GBC *ERBB2* amplifications are more commonly identified [95]. Overall, low numbers of patients with BTC have shown benefit from immunotherapy and NTRK-targeted therapy in clinical trials {Valle, 2021, Biliary tract cancer}. Indeed, the ESMO recommends multigene testing in patients with advanced CCA, in order to detect the full picture of targetable alterations [28]. In this line, a study including 327 patients with BTC revealed that patients receiving genetic alteration-matched targeted therapy in the second line showed better survival than patients without targetable alterations. Particularly, survival was better in patients with ESCAT I-II than the ESCAT III-IV alteration [24]. 

Different studies have assessed the feasibility of ctDNA genetic testing in BTC and compared its performance to tissue genetic testing. Overall, a similar genetic landscape is identified with both approaches, albeit with different concordance levels depending on experiment settings and testing technologies [96,97,98,99,100,101,102,103,104]. The largest study in this regard included 1671 patients and detected ctDNA in 84% of them, of whom 44% harbored targetable alterations. This study highlights a good concordance level between ctDNA and tissue NGS, and the advantage of ctDNA in terms of repeated sampling to follow tumor clonality and arising resistance-mechanisms [99]. Another study analyzed genetic alterations in 121 patients by tissue or ctDNA NGS, showing better survival in patients treated with matched targeted therapies than in patients with un-matched treatment [98]. Moreover, this work reported a better concordance of genetic alterations found in ctDNA with metastatic lesions than with the primary tumor, suggesting that novel alterations may arise in metastasis and that ctDNA analysis may be more informative in the selection of patients for matched targeted therapy. In this regard, a recent study reported a lower failure rate with blood genetic testing than with tissue (15.4% of no detectable alterations in blood vs. 26.8% in tissue), insufficient tumor tissue being the most common cause for tissue testing failure. Importantly, ctDNA-based genetic testing could be an alternative in this situation [102]. Of note, another study focused on early-onset BTC revealed a distinct genetic landscape in patients with early-onset BTC compared to older patients [104]. 

### 5.3. Prognosis and Disease Monitoring

Additionally, ctDNA analysis holds a prognostic value [105,106,107]. Overall, higher VAF of the dominant genetic alteration prior to treatment associates with worse clinical outcomes than PFS or a higher tumor burden [99,103,105,107]. Intriguingly, some other studies report no association between VAF and clinical outcomes [102,106]. A recent study proposes a CNV score based on plasma ctDNA analysis that is able to predict response to immunotherapy in patients with hepatobiliary malignancies. Among patients treated with immune checkpoint inhibitors, those with lower CNV risk scores had longer OS and PFS than those with high CNV risk scores [107].

Finally, the analysis of ctDNA to detect genetic alterations conferring acquired resistance to targeted therapy has also been assessed in BTC. Two studies have monitored resistance mechanisms in FGFR2-positive-patients treated with FGFR2 inhibitors by serial ctDNA analysis. Importantly, ctDNA was able to identify multiple resistance mechanisms not detected by tissue biopsy, even before detection of progression by radiologic imaging [108,109]. Table 3 summarizes key findings and designs of selected studies assessing the role of ctDNA in BTC.

## 6. Pancreatic Ductal Adenocarcinoma (PDAC)

PDAC is an aggressive and fatal malignancy. It ranks seventh among cancer deaths worldwide, with an estimated 495,773 new cases and 466,003 deaths in 2020 [38]. Nevertheless, it is much more common in highly developed countries, in which it will probably surpass breast cancer as the third leading cause of cancer death by 2025 [110].

Around 5–10% of PDAC comprise germline alterations, *BRCA1*/*BRCA2* being the most commonly mutated genes [111,112], thus identifying a subset of patients with a potential benefit of PARP inhibitors [112,113]. Finally, the most common driver mutated genes in PDAC are non-druggable, and chemotherapy constitutes the unique therapeutic approach in the metastatic setting [111,114]. Considering *KRAS* as the most frequently mutated gene [115,116,117,118,119,120,121,122], multigene NGS evaluation could be of interest in those non-mutated cases for targeted treatment considerations [10]. Pan-cancer studies have reported promising results with targeted therapy for PDAC patients with MSI-H and *NTRK* fusions [123,124].

### 6.1. Screening and Diagnosis

Screening for PDAC is not recommended except for those individuals with a family history or presenting several associated conditions [111]. In this sense, due to the unspecific symptoms, PDAC is normally detected in advance stages and only 20% of cases are surgically treatable [114]. The definitive diagnosis needs an invasive procedure ordinarily performed with a fine needle aspiration cytology of the primary tumor, which relies on sufficient tumor tissue for an accurate molecular analysis [111]. The protein cancer antigen (CA) 19.9, which is measured in clinical practice, is commonly increased not only in an advanced stage of disease but also parallel to bilirubin levels, resulting in an unspecific marker with frequent false-positive results and not useful for initial diagnosis [111]. This highlights the importance of finding a reliable marker to detect the disease at early stages. In patients with resectable tumors, the presence of preoperative ctDNA was associated with larger tumor size, lymph node positivity and the presence of microscopic lymphovascular invasion [125]. In this sense, some studies have shown that levels of ctDNA are significantly greater in those patients with PDAC compared with healthy controls [118,126] but not with chronic pancreatitis [119,127]. Moreover, it has been observed that both ctDNA detection and the number of somatic alterations are higher in those patients with metastatic disease in comparison with patients with resectable locally advanced or early-stage tumors [116,121,128]. 

### 6.2. Prognosis

The prognostic value of ctDNA has been assessed in both early and advanced PDAC. Several studies report that the detection of *KRAS* mutations in ctDNA prior to active treatment or surgery associates with worse survival rates, as well as with certain clinico-pathological features. Particularly, *KRAS* mutations in ctDNA associates with tumor location in tail and neck, advanced stages, diagnosis of liver metastasis and high numbers of circulating regulatory T-cells [119,120,122,125,126,127,128,129,130,131,132,133,134,135,136,137,138,139,140]. Interestingly, in some studies, detection of *KRAS* mutations in tissue does not show prognostic significance, while ctDNA-based testing does [120]. Genetic alterations in other genes or ctDNA-related factors may also hold a prognostic value. In this regard, alterations in chromatin-regulating genes are linked to better outcomes [115] while higher ctDNA concentration, *ERBB2* mutations or methylation sites in ctDNA associate with worse outcomes [141,142,143]. Studies that assess the genetic concordance between ctDNA and tissue biopsy obtained varying results [115,119,120,128,129,139,141]. 

### 6.3. Disease Monitoring

Besides its diagnostic properties, ctDNA also serves to monitor patient response in different settings. In resectable PDAC, it has been observed that both neoadjuvant therapy and surgery reduce the levels of ctDNA [125,139], and its detection after surgery could be predictive of an early relapse and worse outcome [115,116,125,134,136,139,144,145,146]. Indeed, some studies, including small subsets of patients, suggest that detection of MRD by ctDNA analysis can anticipate imaging-based diagnosis of relapse [115,116,125], and it also outperforms the predictive capacities of protein markers such as CA 19.9 [125,131,134,139,140,145,146]. Similarly, in metastatic PDAC, ctDNA analysis during chemotherapy serves as a marker to monitor response, as it associates with worse survival [117,118,130,145,147]. In this regard, a recent clinical trial assessed the role of ctDNA to monitor responses to second-line treatment based on enzyme administration, showing that ctDNA evolution after treatment correlated with outcomes [117]. Table 4 summarizes key findings and study design of selected studies assessing the role of ctDNA in PDAC.

## 7. Conclusions and Future Directions

Analyses of ctDNA have emerged as a revolutionary tool for the management of patients with cancer over the past decade. Although the gold standard for tumor histotype characterization and first-line therapeutic approaches relies on the analysis of the tumor tissue, ctDNA constitutes an alternative source of tumor-derived DNA when tumor tissue sampling is challenging. In non-small-cell lung cancer, for instance, ctDNA analysis has become part of the armamentarium for treatment decision-making. Of note, ctDNA is unable to define either the tumor histology or the local microenvironment, but it does challenge the genomic testing. Considering upper gastrointestinal malignancies, several studies have shown the utility of ctDNA analysis, although further prospective validation is needed. Advantages over genetic testing in traditional tissue biopsy include the minimally invasive nature of the test and its ability to reveal a more holistic overview of the tumor genomic landscape, thus covering the temporal and spatial heterogeneity of tumor biology.

Studies in GEC, BTC and PDAC have shown how liquid biopsy can potentially impact early cancer detection and prognostication in all tumor stages and may identify the progression earlier than imaging techniques. Additionally, in the advanced setting, ctDNA analysis serves as a non-invasive tool for genetic testing and targeted therapy selection as well as for monitoring therapy response, being able to detect multiple resistance mechanisms. With such an amount of evidence, data generated adds convincement to researchers to rely on the potential role of liquid biopsy in these lethal tumors. In the era of personalized medicine, ctDNA analysis in liquid biopsy emerges as a highly valuable paradigm. Nevertheless, most of the studies are still observational and include small cohorts of patients. Future, prospective multi-center and interventional studies, carefully designed to base clinical decision-making on ctDNA results, will unveil the real clinical role of ctDNA analysis for upper gastrointestinal tumors and allow its integration into the therapeutic armamentarium.

## Figures and Tables

**Figure 1 cancers-15-01379-f001:**
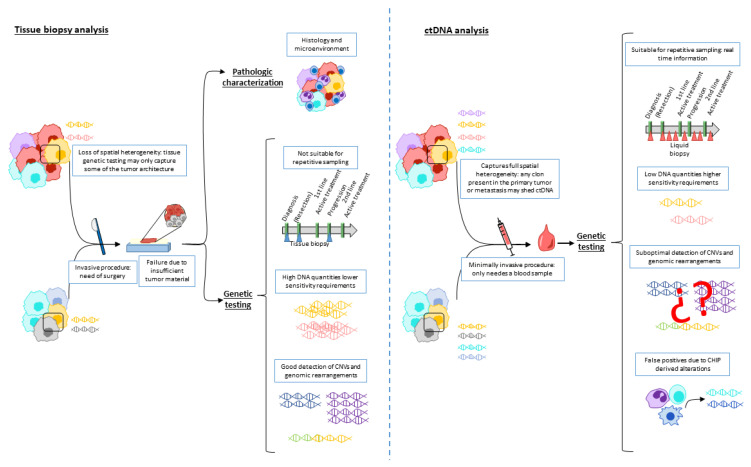
Characteristics of genomic testing in tissue and ctDNA of liquid biopsy. Tissue analysis allows for histotyping and tumor microenvironment characterization. Genetic testing in tissue biopsy (**left** panel) needs an invasive procedure, thus preventing repetitive testing. It may also show high failure rates due to insufficient tumor material and may only represent part of the entire tissue architecture. On the other hand, high DNA quantities are usually extracted, which eases detection of genetic alteration including CNVs and genomic rearrangements. Genetic testing ctDNA (**right** panel) is minimally invasive, allowing repetitive testing, and can cover full tumor heterogeneity. Nevertheless, low DNA quantities require highly sensitive technologies, CNV and genomic rearrangement detection is still unsatisfactory, and CHIP may lead to false positives. Abbreviations: CNV: copy number variation; ctDNA: circulating tumoral DNA; CHIP: clonal hematopoiesis of indeterminate potential.

**Table 1 cancers-15-01379-t001:** Key findings and study design of main studies assessing the role of ctDNA in different types of cancer.

Study-Authors	Study Population	Collection of Blood Sample	Technology	Key Findings
Bettegowda et al. (2014) [11]	21 GEC; 155 PDAC (out of 410 cancer pts)	ns	PCR-based technology (EGFR pathway genes)	- ctDNA detectable in >75% of advanced stage, for localized tumors, ctDNA detectable in 57% GEC and 48% PDAC.
Chen et al. (2020) [30]	217 GEC; 52 liver (out of 414 cancers, 191 pre-diagnosis; 223 post-diagnosis and 414 healthy controls)	Basal prior to therapy	Methylation assay	- High sensitivity to detect cancer in post-diagnosis samples (96.5%) and capable to detect cancer even 4 years before diagnosis.
Klein et al. (2021) [32]	130 GEC, 63 hepatobiliary and GBC, 135 PDAC (out of 2823 cancer pts and 1254 non-cancer controls) (from CCGA trial)	Basal prior to therapy	Methylation assay	- High specificity (99.5%) and sensitivity (51.5%) for early cancer detection - Sensitivity increased with advanced stages.
Zhang et al. (2020) [35]	GEC 48; liver 38; PDAC 42 (out of 978 advanced cancer patients) (from: Study 1108, ATLANTIC and Study 10 trials)	Basal prior to therapy and during therapy (6–8 weeks after initiation)	NGS (73 genes. Guardant360)	- High VAF in the basal sample associates with worse clinical outcomes. - VAF reduction during treatment associates with PFS, OS and ORR
Parikh et al. (2019) [37]	6 GEC; 4 BTC (out of 42 advanced cancer pts with acquired resistance to targeted therapy)	Post-progression to targeted therapy.	NGS (73 genes. Guardant360)	- ctDNA analysis detect resistance mechanisms not detectable in tissue biopsies in 78% of the cases. - Acquired resistance mechanisms are highly heterogeneous and polyclonal.
Nakamura et al. (2020) [34]	260 GEC; 188 CCA; 363 PDAC (out of 1687 advanced cancer pts)	Basal prior to therapy	NGS (73 genes. Guardant360)	- ctDNA based genetic testing for clinical trial enrolment decreases screening time and improves enrolment rate, without impacting clinical outcomes. - ctDNA reveals novel targetable oncogenic alterations.

Abbreviations: BTC, biliary tract cancer; CCA, cholangiocarcinoma: ctDNA, circulating tumor DNA; EGFR, epidermal growth factor receptor; GEC, gastroesophageal cancer; NGS, next-generation sequencing; ORR, objective response rate; PDAC, pancreatic ductal adenocarcinoma; PFS, progression-free survival; pts, patients; VAF, tumor somatic variant allelic frequency.

**Table 2 cancers-15-01379-t002:** Key findings and study design of main studies assessing the role of ctDNA in GEC.

Study	Study Cohort	Technology	Key Findings
Lan et al. (2017) [33]	428 GEC (out of 855 gastrointestinal cancer pts and 95 healthy donors)	qPCR	- Higher levels of cfDNA in GEC compared with non-cancer donors. - ctDNA in plasma after surgery positively correlate with recurrence, better than serum biomarkers (CEA).
Catenacci et al. (2021) [45]	15 *HER2*-amplified GEC (out of 80 GEC) (from PANGEA trial)	NGS (73 genes. Guardant360)	- HER2 baseline spatial heterogeneity. - HER2 conversion over time to anti-HER2 therapies.
Qian et al. (2017) [49]	124 GEC; 64 benign gastric disease; 92 healthy donors	Alu-based bDNA	- GEC present higher levels of cfDNA, compared with non-cancer donors. - cfDNA more sensible than serum markers (CEA, CA 19.9, CA 72.4) for detecting early GEC.
Maron et al. (2019) [52]	1630 GEC	NGS (73 genes. Guardant360)	- Detectable ctDNA in the postoperative period correlates with inferior DFS. - Genomic alterations more frequently detected in ctDNA-NGS than in tissue-NGS (temporospatial heterogeneity). - Temporal changes in VAF correlate with treatment prognosis. - ctDNA-NGS *HER2* amplifications in 11.3% (discordance with tumor-NGS); changes correlated with response to anti-HER2 targeted therapies.
Ococks et al. (2021) [53]	97 GEC	NGS (77 genes. Avenio ctDNA Expanded)	- ctDNA in plasma after surgery positively correlates with recurrence.
Yang et al. (2020) [54]	46 GEC	NGS (1021 genes. Custom)	- ctDNA in plasma after surgery positively correlates with recurrence.
Openshaw et al. (2020) [55]	40 GEC	ddPCR (SNVs) qPCR (SCNAs)	- ctDNA in plasma after surgery positively correlates with recurrence. - High ctDNA levels at diagnosis of metastatic disease predicted poor survival
Kim et al. (2019) [56]	25 GEC	WGS (Sanger sequencing)	- ctDNA in plasma after surgery positively correlates with recurrence.
Wo et al. (2021) [57]	21 GEC	ddPCR	- ctDNA in plasma after surgery positively correlates with recurrence.
Fedyanin et al. (2020) [58]	42 GEC	ddPCR	- ctDNA in plasma after surgery positively correlates with recurrence.
Leal et al. (2020) [59]	50 GEC (from CRITICS trial)	NGS (58 genes. Agilent SureSelect)	- ctDNA in plasma after surgery positively correlates with recurrence.
Huffman et al. (2022) [60]	295 GEC	Personalized multiplex PCR-based NGS assay	- ctDNA in plasma after surgery positively correlates with recurrence.
Bang et al. (2015) [62]	71 FGFR2b-positive GEC (from SHINE trial)	Nanostring *(FGFR2* gene expression)	- *FGFR2* CNV predicts response to anti-FGFR2 inhibitors.
Jin et al. (2020) [65]	46 GEC	NGS (425-genes)	- Temporal changes in VAF correlate with PFS and response rate.
Davidson et al. (2019) [66]	30 GEC	DNA SCNA sLow-coverage WGS	- ctDNA levels prior chemotherapy positively correlate with survival - SCNAs profiles changed during chemotherapy (clonal evolution).
Kim et al. (2018) [67]	61 GEC	NGS (73 genes. Guardant360)	- Changes in ctDNA levels predict response and PFS to immunotherapy (decrease of ctDNA levels correlated with improved outcomes).
Chen et al. (2019) [68]	55 GEC	WGS (SCNAs)	- Copy number instability score correlate with response to treatment (decrease of the score in pts responding to treatment).
Wang et al. (2018) [69]	24 HER2-positive GEC	NGS (416 genes)—*HER2* SCNAs	- High concordance of *HER2*-tissue amplification and *HER2*-blood SCNAs. - High *HER2* SCNA in pts with innate resistance to anti-HER2 therapies. - *HER2* SCNAs decrease in pts wit acquired resistance to anti-HER2 therapies. - Feasibility to identify possible mutated genes that confer resistance to anti-HER2 therapies.
Wang et al. (2018) [70]	56 HER2-positive GEC	NGS (416 genes. Illumina HiSeq 2500)	- High concordance with tissue analyses. - Correlation of HER2 copy number with response to anti-HER2 therapies.
Shoda et al. (2017) [71]	15 HER2-positive GEC	ddPCR-based *HER2* copy number	- High concordance with tissue analyses; correlation of *HER2* copy number with response to anti-HER2 therapies
Kim et al. (2018) [72]	32 HER2-positive GEC (phase II study)	NGS (73 genes. Guardant360)	- Correlation between plasma *HER2* copy number and response to anti-HER2 therapies. - Serial ctDNA sequencing demonstrates tumor evolution and change in genomic profile.
Qiu et al. (2020) [73]	140 EBV-positive GEC (out of 2760 GEC)	Plasma EBV-DNA load	- Only 52.1% of EBV-positive GC pts have detectable plasma EBV-DNA - Plasma EBV-DNA levels positively correlate with disease stage. - Dynamic changes on plasma EBV-DNA levels correlates with treatment response.
Wainberg et al. (2022) [75]	155 FGFR2b-positive GEC (from FIGHT trial)	NGS for *FGFR2* amplification; Personal Genome Diagnostics PGDx elio plasma resolve	- *FGFR2* amplification is biomarker of response to FGFR2 inhibitors.
Smyth et al. (2020) [76]	354 EGFR-positive GEC (from REAL3 trial)	ddPCR (*EGFR*)	- High concordance between *EGFR* copy number in tissue and in liquid biopsies (95%). - Plasma *EGFR* copy number is a (bad) prognostic biomarker, although does not correlate with response.

Abbreviations: CA, cancer antigen; bDNA, branched DNA; CEA, carcinoembryonic antigen; cfDNA, cell free DNA; ctDNA, circulating tumor DNA; ddPCR, droplet-based digital polymerase chain reaction (PCR); DFS, disease free survival; EBV, Epstein Barr Virus; GEC, gastroesophageal cancer; NGS, next-generation sequencing; PFS, progression free survival; pts, patients; qPCR, quantitative PCR; SCNAs, somatic copy number alterations; SNVs, somatic single-nucleotide variants; VAF, tumor somatic variant allelic frequency; WGS, whole-genome sequencing.

**Table 3 cancers-15-01379-t003:** Key findings and study design of main studies assessing the role of ctDNA in BTC.

Study	Study Cohort	Timing of Sample Collection	Technology	Key Findings
Wang et al. (2021) [81]	29 BTC; 18 benign biliary lesions	Basal at diagnosis	NGS (low coverage WGS)	- ctDNA CNV assay better diagnostic capacity than CA 19.9, combination of ctDNA CNV and CA 19.9 best diagnostic capacity. - Higher CNV burden associated with worse overall survival.
Wasenang et al. (2019) [82]	40 early stage CCA; 40 benign biliary lesions	Basal at diagnosis	Methylation assay	- Combination of two methylation markers in ctDNA show good diagnostic capacities.
Kumari et al. (2017) [83]	34 GBC; 39 Controls	Basal at diagnosis	PCR	- cfDNA shows good diagnostic capacities to distinguish GBC from healthy controls and from benign biliary lesions (AUC 0.983).
Kumari et al. (2019) [84]	60 GBC; 36 controls	Basal at diagnosis	PCR and methylation assay	- cfDNA integrity value increased in GBC compared to controls; no differences in global methylation levels. - cfDNA integrity good diagnostic.
Han et al. (2021) [85]	42 BTC	Basal prior to therapy	ddPCR for KRAS	- Bile ctDNA analysis feasible, better concordance between bile and tissue (80%) than between plasma and tissue (42.9%).
Arechederra et al. (2022) [86]	68 BTC or PDAC and benign biliary lesions	Basal prior to therapy at diagnosis	NGS (52 genes. Oncomine focus)	- Bile ctDNA feasible, more mutations detected in bile ctDNA than in tissue or plasma. - Better diagnostic capacity than clinicopathological evaluation.
Shen et al. (2019) [87]	10 stage II, III and IV BTC	Basal prior to therapy	NGS (150 genes. Custom)	- Bile ctDNA feasible, bile and tissue DNA similar alterations.
Kinugasa et al. (2018) [88]	30 stage II, III and IV GBC	Basal prior to therapy	NGS (48 genes. Custom)	- Bile ctDNA feasible, Correlation with tissue 85.7% - Better diagnostic capacity than cytology evaluation
Gou et al. (2021) [89]	28 BTC	Basal prior to therapy	NGS (520 genes. Custom)	- Bile ctDNA feasible, better correspondence between bile ctDNA with tissue than plasma with tissue. - Bile ctDNA + CA 19.9 best diagnostic approach.
Zill et al. (2015) [96]	26 stage III and IV BTC/PDAC *(18 PDAC; 8 BTC)*	Basal prior to therapy	NGS (54 genes. Guardant)	- ctDNA analysis is feasible for cancer genotyping, high level of concordance between tissue and ctDNA. - Tissue genetic testing failed in 35% patients; ctDNA was able to identify mutations in 78% of those.
Kim et al. (2015) [97]	38 metastatic BTC	ns	ddPCR (*PIK3CA* mutations)	- ctDNA analysis is feasible for cancer genotyping, total concordance between ctDNA and tissue.
Okamura et al. (2021) [98]	121 advanced BTC	ns	NGS(68–73 genes. Guardant)	- ctDNA and tissue genetic testing are feasible for tumor genotyping. - ctDNA concordance assessed for specific genes, better concordance of ctDNA with metastasis than primary tumor.
Berchuck et al. (2022) [99]	1671 advanced BTC-(different subcohorts for each analysis)	General ns; a subcohort of Basal prior to therapy	NGS (70–73 genes. Guardant)	- ctDNA analysis is feasible for cancer genotyping, detects alterations in 84% of the patients and targetable alterations in 44%. - High concordance between tissue and ctDNA for *IDH1* and *BRAF* mutations (87% and 100%), low for *FGFR2* fusions (18%). - High VAF in pre-treatment ctDNA associates with worse outcomes.
Chen et al. (2021) [100]	154 advanced BTC	ns	NGS (150 genes. Custom)	- ctDNA analysis is feasible for cancer genotyping; 94.8% of the patients showed at least one alteration. - Most frequently altered genes similar in tissue and ctDNA.
Csoma et al. (2022) [101]	25 BTC	24 months after initial tissue biopsy or surgical resection	NGS (67 genes. Archer)	- ctDNA analysis is feasible for cancer genotyping; 84% of the patients show some SNV. - Tumor variant burden (number of variants per sample) similar in tissue and ctDNA.
Lamarca et al. (2020) [102]	104 advanced BTC	ns	NGS (70 genes. FoundationOne.)	- ctDNA and tissue genetic analysis is feasible for tumor genotyping, high concordance with tissue in *FGFR2* and *IDH1* genes (100%). - Failure rate higher in tissue than ctDNA (26.8 vs. 15.4%).
Ettrich et al. (2019) [103]	24 stage III and IV BTC	Basal prior to therapy and serial sampling in a subcohort	NGS (15 genes panel. Custom/CeGaT. 710 gene panel)	- ctDNA analysis is feasible for tumor genotyping, tissue and ctDNA concordance overall 74% and 92% for intrahepatic CCA. - VAF in ctDNA associates with poorer outcomes. - Chemotherapy changes the ctDNA mutational landscape.
Mody et al. (2019) [104]	130 advanced BTC	Any time point (ns)	NGS (73 genes. Guardant)	- ctDNA analysis is feasible for cancer genotyping; 55% of the patients showed at least one targetable alteration.
Uson Junior et al. (2022) [105]	67 metastatic BTC with (1st line with platinum chemotherapy)	Basal prior to therapy	NGS (73 genes. Guardant)	- ctDNA holds prognostic value; high VAF in ctDNA is associated with worse overall survival.
Lapin et al. (2022) [106]	31 metastatic BTC (IDH matched therapy)	Basal prior to therapy, during therapy, at progression	NGS (73 genes. Guardant)/ddPCR	- ctDNA holds prognostic value; lower variant allele frequency at baseline correlates with longer time to treatment failure. - Emergence of novel mutations at progression.
Yang et al. (2021) [107]	187 advanced hepatobiliary 47 HCC, 115 BTC, 5 mixed	Basal prior to therapy (ns)	NGS including SNVs and CNVs	- ctDNA holds prognostic value; lower CNV risk score measured in ctDNA at baseline improved clinical outcome to ICI therapy.
Goyal et al. (2017) [108]	3 advanced BTC (FGFR2 matched therapy)	Serial sampling during treatment and at progression	NGS (Guardant. 70 genes)	- ctDNA is able to detect multiple resistance mechanisms not detectable in a single tissue biopsy.
Varghese et al. (2021) [109]	8 advanced BTC (*FGFR2* matched therapy)	Serial sampling during treatment and at progression	NGS (129 genes. Custom)	- ctDNA is able to detect multiple resistance mechanisms.

Abbreviations: AUC, area under the curve; BTC, biliary tract cancer; CA, cancer antigen; CCA, cholangiocarcinoma; cfDNA, circulating free DNA; CNV, copy number variations; ctDNA, circulating tumor DNA; GBC, gallbladder cancer; HCC, hepatocellular carcinoma; ICI, immune checkpoint inhibitor; IDH, isocitrate dehydrogenase; ddPCR, droplet-based digital polymerase chain reaction (PCR); NGS, next-generation sequencing; ns, not specified; PDAC, pancreatic ductal adenocarcinoma; SNV, single-nucleotide variants; VAF, tumor somatic variant allelic frequency.

**Table 4 cancers-15-01379-t004:** Key findings and study design of main studies assessing the role of ctDNA in PDAC.

Study-Authors	Study Cohort	Collection of Blood Sample	Technology	Key Findings
Sausen et al. (2015) [115]	77 stage II (10 pts monitoring)	At diagnosis	NGS (whole exome and 116 genes—custom)/ddPCR (*KRAS*, *BRAP* and *PIK3CA*)	- ctDNA detection at diagnosis and post-surgery, predictor of relapse. - ctDNA detected post-surgery ≈ 6.5 mo anticipates recurrence detection by imaging.
Pietrasz et al. (2017) [116]	135 (8 pts monitoring)	Prior first-line chemotherapy and/or after surgery	NGS (22 genes. Ion AmpliSeq^TM^ Colon and Lung Cancer Panel v2)/ddPCR (*KRAS*)	- ctDNA associated with grade of tumor differentiation and greater in metastatic (74.7% metastatic vs. 16.6% locally advanced vs. 19% resectable). No correlation with number of metastatic sites. - ctDNA detection correlated with poor OS in advanced stages (6.5 vs. 19 mo). Post-surgery, negative ctDNA associated with longer DFS (17.6 vs. 4.6 mo) and OS (32.2 vs. 19.3 mo). - ctDNA can anticipate progression detection to imagining-based techniques. ctDNA correlated with chemotherapy response.
Bachet et al. (2020) [117]	113 metastatic (88 pts first and second cycle) (from phase II trial)	Prior first, second and third cycles	NGS (22 genes. Ion AmpliSeq^TM^ Colon and Lung Cancer Panel v2)	- ctDNA+ at baseline associated with shorter OS (4.6 mo vs. 8.8 mo) and PFS (1.6 vs. 3.3), but not with ORR. Higher tertile of VAF prognostic of OS and PFS. - During monitoring: ctDNA change associated with ORR; disease control better in ctDNA negative and with decrease maximal VAF at baseline.
Kinugasa et al. (2015) [120]	75 (discovery cohort with matched tissue) and 66 (validation cohort); 20 healthy and 20 CP	Prior to therapy	ddPCR (*KRAS*-mut)	- Concordance ≈ 77.3% between tissue and ctDNA. *KRAS*-mut in greater proportion in cases (63–55%) vs. controls (5%) and CP patients (20%). - Survival time was shorter in *KRAS*-mut vs. *KRAS*-wild type in serum. Stage of disease, *KRAS*-mut and p.G12V mutation significant factors for survival in both cohorts.
Groot et al. (2019) [125]	59 resectable	Prior and post-surgery	ddPCR (*KRAS*-mut)	- ctDNA detection prior-surgery associated with clinico-pathological parameters, decreased median RFS and OS. - ctDNA detection prior adjuvant were likely to relapse and had reduced RFS (5 vs. 15 mo). ctDNA+ preceded detection by imaging in 81%. No association between ctDNA and CA 19.9.
Cheng et al. (2020) [138]	210 stage III and IV	Prior treatment	ddPCR (*KRAS*-mut)	- *KRAS* p.G12V mutation associated with Tregs high levels, no relationship with KRAS p.G12D. - TNM stage, chemotherapy, Tregs, CA 19.9, CA 125 and KRAS associated with OS. Patients with—*KRAS* p.G12V mutation and high Tregs had worse survival (4.5 vs 8.5 mo).
Patel et al. (2019) [141]	112 advance (14 pts resectable tumor)	ns for advanced; before or after surgery for resectable patients	NGS	- ctDNA detectable in 75% of advanced vs. 50% resectable tumors. - Better concordance: between ctDNA and metastatic tissue (72%) than primary tumor (39%). - *KRAS*-mut and higher levels of ctDNA associated with worst OS (7.4 vs. 11.4 mo and 6.3 vs. 11.7 mo). - 90% of alterations in ctDNA potentially targetable by FDA-approved agents. 73% from advanced had at least one actionable alteration.
Pietrasz et al. (2022) [142]	110 locally advanced and metastatic (test cohort) and 255 metastatic (validation cohort) (from PRODIGE 35 and 37 trials)	Prior first-line chemotherapy (test cohort) or at inclusion (validation cohort)	NGS (22 genes. Ion AmpliSeq^TM^ Colon and Lung Cancer Panel v2)/methylation-ddPCR (HOXD8 and POU4F1)	- Methylation at *HOXD8* and *POU41* genes could be marker of ctDNA. - ctDNA detection in 63.7% from test cohort (95% with liver metastasis vs 56% without metastasis). - ctDNA detection associated with worse OS and PFS.
Nakano et al. (2018) [144]	45 stage I-II	Prior-surgery and prior-discharge	Clamp PCR (*KRAS*-mut)	- *KRAS*-mut+ after surgery (not before) had shorter DFS and OS vs. *KRAS*-wild type. *KRAS*-mut associated with early recurrence. - Change from *KRAS*-wild type to *KRAS*-mut after surgery prognostic factor for poor OS.
Watanabe et al. (2019) [145]	78 localized, metastatic and recurrent	Prior first-line chemotherapy; prior and post-surgery (≥3 serial samples)	ddPCR (*KRAS*-mut)/RASKET (MEBGEN kit)	- Concordance *KRAS*-mut: 94.7% tumor center vs invasion and 90.9% primary tumor vs metastasis. - No association *KRAS*-mut and CA 19.9 levels before surgery with RFS. - *KRAS*-mut+ pre-chemotherapy had decreased OS. CA 19.9 no related. - *KRAS*-mut emergence associated with worse OS (1-year post-surgery) and PFS (within 6 mo of chemotherapy/4.8 vs. 14.9 mo). No changes in CA 19.9.
Kruger et al. (2018) [146]	54 locally advanced and metastatic	Prior chemotherapy; weekly during 2 mo; and at time of radiological staging and treatment	BEAMing PCR (*KRAS*-mut)	- Concordance between plasma and tissue ≈ 75% and only in metastatic ≈ 79%. - *KRAS*-mut+ in plasma at baseline correlated with less OS and PFS. - Changes in *KRAS*-mut were more pronounced and rapid than changes in tumor markers during monitoring. - At 1st month of chemotherapy, correlation between changes in ctDNA and response to treatment, no correlation with protein-based markers.
Del Re et al. (2017) [147]	27 locally advanced and metastatic (25 with monitoring)	Prior and post 15th day chemotherapy; and at first radiological test	ddPCR (*KRAS*-mut)	- No differences in ctDNA metastatic vs. locally advanced, gender, age, site. - At 15th day, patients with increase *KRAS*-mut had less PFS and disease progression. - *KRAS*-mut increase associated with less OS. Early mutation does not correlate with tumor response.

Abbreviations: CA, cancer antigen; cfDNA, cell-free DNA; ctDNA, circulating tumor DNA; CP, chronic pancreatitis; ddPCR, droplet-based digital polymerase chain reaction (PCR); DFS; disease-free survival; *KRAS*-mut, *KRAS* gene mutations; MAF, mutant allelic frequency; mo, month; MST, median survival time; NGS, next-generation sequencing; ns, not specified; ORR, objective response rate; OS, overall survival; PDAC, pancreatic ductal adenocarcinoma; PFS, progression-free survival; RFS, relapse-free survival; Tregs, regulatory T-cells.

## Data Availability

Data supporting reported results are the references described above.

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
