# Peer review of "State of the Art: ctDNA in Upper Gastrointestinal Malignancies"

_cancers, 2023, doi:10.3390/cancers15051379_

Round 1

Reviewer 1 Report

I thank the Editor for having submitted this interesting paper to me.

The text is well written and well organized, pleasant and smooth to read. The topic is timely and the Authors provide a detailed and updated literature review.

However, there are important points to clarify.

1- The liquid biopsy (ctDNA) is unable to define the histotype of the neoplasm according to WHO classification (Squamous cell carcinoma, adenocarcinoma, neuroendocrine carcinoma, GIST, others...). Please, it is important to remember that histotype drives the first-line therapy and prognosis. Moreover, with small amount of ctDNA it is essential to need neoplastic histotype to indagate appropriate gene mutations (i.e. Kit in GIST insite of EGFR for adenocarcinoma).

2- As demostrated in Lung adenocarcinomas in progression under therapy with TK inhibitors, the liquid biopsy (ctDNA) is unable to identify any histotype shift (i.e. in small cell lung cancer), needing another therapeutic approach. In this context, ctDNA is inappropriate.

3- The liquid biopsy (ctDNA) is unable to give information on the tumor microenvironment to plan a therapy with Immune chech point inibithors.

Please, better define this topic in each paragraphs and add a Table describing strengths and limitations of ctDNA. Finally, add this topic in the Conclusion.

Please, find the paper reviesd enclosed.

Author Response

We thank the Reviewer for the comments and suggestions, which we have addressed accordingly (see attached document answering to all of the questioned points).

Reviewer 2 Report

I would like to congratulate the authors of the current review

I have just two comments to make:

1) The authors are using quite often the word “deadly” for a specific type of tumor. I would prefer rather than using tis word, to utilize the phrase “with poor prognosis” or “with bad prognosis”.

2)I would also propose a different outline of the first part of the manuscript witch in my opinion makes more sense:

I would start again the manuscript with the “introduction” however I would insert there the lines 81-90 and 32-36 of the current version of the manuscript.

I would then add the Section “Technical aspects of ct-DNA analysis” and I would use there the lines

110-133, 90-98 and figure 1 of the current version of the manuscript.

I would then add a section with a title such as “General aspects of clinical applicability of ctDNA in upper gastrointestinal malignancies” and I would use there the lines of the current version of the manuscript 37-79.

I would then continue with the outline as it is now “clinical application of ctDNA analysis in GEC, BTC and PDAC” etc.

Author Response

(The authors gave the same response as above.)

Round 2

Reviewer 1 Report

The paper is improved.

Congratulations.